# A flexible cross-platform single-cell data processing pipeline

Kai Battenberg [1,2,5], S. Thomas Kelly [1,5], Radu Abu Ras[1,3], Nicola A. Hetherington[4], Makoto Hayashi [2] & Aki Minoda [1,4] ✉

Single-cell RNA-sequencing analysis to quantify the RNA molecules in individual cells has become popular, as it can obtain a large amount of information from each experiment. We introduce UniverSC (https://github.com/minoda-lab/universc), a universal single-cell RNA-seq data processing tool that supports any unique molecular identifier-based platform. Our command-line tool, docker image, and containerised graphical application enables consistent and comprehensive integration, comparison, and evaluation across data generated from a wide range of platforms. We also provide a cross-platform application to run UniverSC via a graphical user interface, available for macOS, Windows, and Linux Ubuntu, negating one of the bottlenecks with single-cell RNA-seq analysis that is data processing for researchers who are not bioinformatically proficient.

Single-cell genomics technologies have driven a recent surge in studies of cellular heterogeneity. Cell throughput has increased over the years and current single-cell RNA-seq (scRNA-seq) technologies can routinely generate data for thousands to hundreds of thousands of cells in a single experiment, some of which are commercially available. This increase in throughput has made it possible for researchers to apply scRNA-seq to a whole range of tissues as well as whole organisms[1–3]. It is expected that scRNA-seq will become more accurate, more reliable, and cost less per cell, becoming feasible for a wide range of studies as the technology matures[4]. However, there is still a bottleneck in the ability of biologists to process the data upon generating the data. Furthermore, with mounting scRNA-seq datasets generated through different platforms deposited by the labs globally, a unified tool is needed for the integration of many dispersed publicly available datasets by processing the data in the same manner and parameters.

In this work, we have developed a data processing tool called UniverSC that will aid in democratising single-cell RNA-seq technology by providing the community, especially biologists who are not familiar with bioinformatics, with a user-friendly tool to process scRNA-seq data generated by any platform.

## Results

### UniverSC runs Cell Ranger on scRNA-seq data of any platform

A common workflow for many of the scRNA-seq technologies involves capturing individual cells, either in gel emulsion with beads or in wells, followed by the addition of a unique molecular identifier (UMI) to RNA molecules, which makes it quantitative. Leveraging the observation that most scRNA-seq technologies utilise the same concept of cell barcodes and UMIs, we developed UniverSC; a shell utility that operates as a wrapper for Cell Ranger (10x Genomics) that can handle datasets generated by a wide range of single-cell technologies. Cell Ranger was chosen as a unifying pipeline for several reasons: 1) it is optimised to run in parallel on a cluster, 2) many labs working on single-cell analysis are likely to already be familiar with the outputs, 3) many tools have already been released for downstream analysis of the output format due to its popularity, 4) the rich summary information and post-processing is useful for further optimisation and troubleshooting if necessary, and 5) the latest open-source release (version 3.0.2) has been optimised further by adapting open-source techniques, such as the third-party EmptyDrops algorithm[5] for cell calling or filtering, which does not assume thresholds specific for the Chromium platform (10x Genomics).

[1]Center for Integrative Medical Sciences, RIKEN, Yokohama, Japan. [2]Center for Sustainable Resource Science, RIKEN, Yokohama, Japan. [3]Faculty of Automatics, Computers and Electronics, University of Craiova, Craiova, Romania. [4]Department of Cell Biology, Faculty of Science, Radboud Institute for Molecular Life Sciences, Radboud University, Nijmegen, The Netherlands. [5]These authors contributed equally: Kai Battenberg, S. Thomas Kelly. ✉e-mail: UniverSC@minodalab.org

UniverSC, which is freely available at GitHub and at DockerHub, can be run on any Unix-based system with the command-line interface. It can also be run on Ubuntu, MacOS, and Windows with a graphical user-interface (GUI), eliminating the need to install or configure separate pipelines for each platform. GUI comes with a function to show the command used for each run, as well as the function to generate reference files. Conceptually, UniverSC carries out its entire process in seven steps (Fig. 1). Given a set of paired-end sequence files in FASTQ format (R1 and R2), a genome reference (as required by Cell Ranger), and the name of the selected technology, UniverSC reformats the whitelist barcodes and sequence files to fit what is expected by Cell Ranger. Additionally, UniverSC provides a file with summary statistics, including the mapping rate, assigned/mapped read counts and UMI counts for each barcode, and averages for the filtered cells. Sequence trimming based on adapter contamination or sequencing quality is not included in the pipeline and no trimming is required to pass files to UniverSC. However, trimming is highly recommended, particularly on R2 files from Illumina platforms, as this generally improves the mapping quality. This requires careful data handling to ensure that all Read 1 and Read 2 are strictly in pairs while only trimming Read 2. We provide a script for convenience that filters Read 1 and Read 2 by the quality scores of Read 2 and avoids mismatching cell barcodes. In principle, UniverSC can be run on any droplet-based or well-based technology (see the software documentation and Table 1 for more details). Settings can also be restored to run on Chromium samples as changes made to the Cell Ranger installation by UniverSC are reversible.

The current release of UniverSC has pre-set parameters for 40 different technologies (Table 1). Further technologies can be used with custom input parameters for any barcode and UMI lengths or by requesting a feature to be added to the GitHub repository. Testing datasets for the following settings are provided: Chromium version 2 and 3 (default), Drop-seq, ICELL8, inDrops-v3, SCI-RNA-Seq, and SmartSeq3.

## UniverSC enables cross-platform single-cell data integration

We demonstrate how our method compares to other data processing pipelines using published datasets. Drop-seq is an example of a droplet-based single-cell technology that does not have known barcodes[6], thus a whitelist of permutations was generated for compatibility. ICELL8 is a well-based technology that has a known barcode whitelist and allows selecting subsets of wells by known barcodes[7]. SmartSeq3 is also a well-based technology that utilises dual indexing and full-length RNA-sequencing[8]. Together with Chromium, these represent several different classes of technologies with different configurations for processing cell barcodes. To assess the degree of similarity between UniverSC and the pipelines for these 4 technologies (Chromium, Drop-seq, ICELL8, and SmartSeq3), both UniverSC and the pipeline used in the original publication of the technique were run on datasets of human cell lines. Specifically, the following pipelines were compared to UniverSC: Cell Ranger (version 3.0.2) (10x Genomics) for Chromium data, dropSeqPipe (version 0.6)[9] for Drop-seq data, CogentAP (version 1.0) (Takara Bio Inc.) for ICELL8 data, and zUMIs (version 2.9.7)[10] for SmartSeq3 data. Our results show high correlation between the gene-barcode matrices (GBMs) generated by UniverSC and the coupled pipelines (identical ($r = 1$) with Cell Ranger 3.0.2 and 0.94 or higher in the three other sets of GBMs, Fig. 2). Correspondingly, clustering results were highly similar based on the high Adjusted Rand Index (ARI) (1 for Chromium, 0.78 for Drop-seq, 0.87 for ICELL8, and 0.78 for SmartSeq3 data, Fig. 2). In the case of UniverSC compared to zUMIs, we do not see a 1-to-1 relationship in UMI counts, despite having a high correlation and a high ARI. This is likely due to the differences in data handling between the two pipelines. While UniverSC discards all multi-mapping reads for UMI counting (function of Cell Ranger), zUMIs includes primary alignments of multimapping reads, leading zUMIs to have a higher UMI count compared to UniverSC. However, the ARI value upon clustering remains high (Fig. 2).

We also demonstrate how applying UniverSC to all datasets from different platforms compares to applying separate pipelines for each technology during data integration. We used published mouse primary cell data from a study benchmarking different scRNA-seq platforms[11]. The Chromium dataset was used as reference and the SmartSeq2 dataset integrated generally well regardless of what pipeline was used for processing (Fig. 3A). However in comparison, processing the SmartSeq2 dataset via UniverSC (and thereby applying a single pipeline to all datasets) resulted in a lower kBET[12] (0.06 compared to 0.11) and a higher Silhouette score[13] (0.43 compared to 0.36) (Fig. 3B, C). This suggests that the batch effect was better removed (based on kBET) and the clusters were more distinct (based on Silhouette score) by UniverSC. A drastic impact was certainly not expected given the high level of correlation between the outputs of UniverSC and various other pipelines tested as above, as well as the fact that all pipelines work under a similar framework. Nevertheless, we demonstrate measurable improvements in data integration by applying UniverSC for all samples, compared to applying separate pipelines on datasets generated by different platforms.

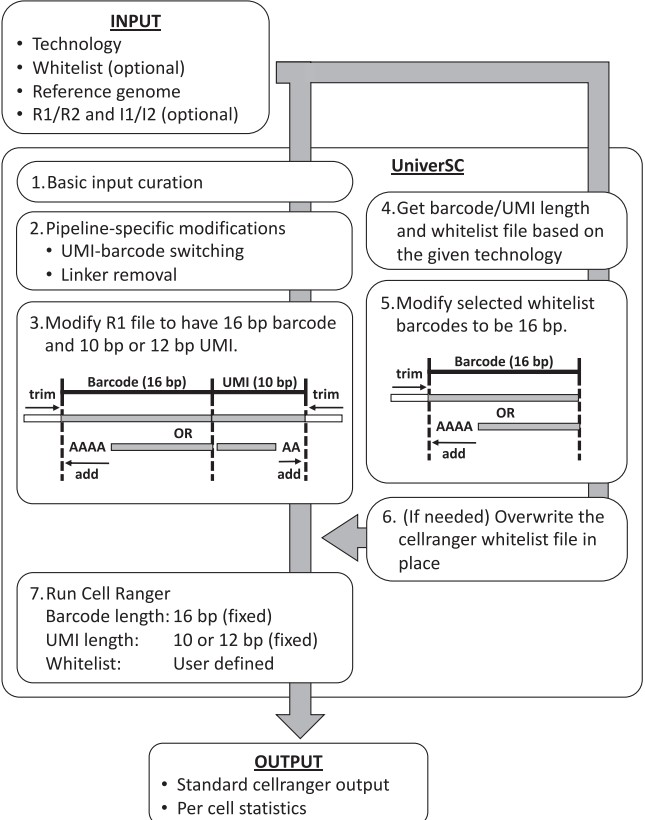

**Fig. 1 | Overview of UniverSC.** Given a pair of FASTQ files (R1 and R2), a genome reference (as required by Cell Ranger), and the name of the technology, UniverSC first runs a basic input curation (step-1). The curated input files are then adjusted for pipeline-specific modification (step-2) and subsequently reformatted to match the expected barcode and UMI lengths (step-3). In parallel, the barcode whitelist suited for the technology (if unspecified by the user) is determined (step-4), and the whitelist barcodes are modified to 16 bp (step-5). If the selected whitelist is different from the whitelist in place for Cell Ranger at the moment, the whitelist is replaced (step-6). Finally, the modified sample data is processed by Cell Ranger against the modified whitelist (step-7) to generate a standard output along with a summary file with per cell statistics.

**Table 1 | Technologies currently available and settings used by UniverSC**

| Parameter value | Technology [Platform, Vendor] | Barcode length[a] | UMI length | Reference |
|---|---|---|---|---|
| 10x-v1 | 10x (version 1) [Chromium, 10x Genomics] | 14 | 10 | [16b] |
| 10x-v2 (or 10x) | 10x (version 2) [Chromium, 10x Genomics] | 16 | 10 | [16b] |
| 10x-v3 (or 10x) | 10x (version 3) [Chromium, 10x Genomics] | 16 | 12 | |
| bravo | HyperCap [Bravo B, Agilent] | 16 | N/A | |
| bd-rhapsody | BD Rhapsody [BD Rhapsody, BD] | 27 | 8 | |
| fluidigm-c1 | C1 [C1, Fluidigm] | 16 | N/A | |
| c1-cage | C1 [Instrument C1] | 16 | N/A | [20] |
| c1-ramda-seq | C1 [Instrument: C1] | 16 | N/A | [21] |
| celseq | CEL-Seq | 8 | 4 | [22,23] |
| celseq2[c] | CEL-Seq2 | 6 | 6 | [23,24] |
| dropseq | Drop-seq | 12 | 8 | [6b] |
| icell8-v2 | ICELL8 (version 2) [ICELL8, Takara Bio] | 11 | N/A | [7b] |
| icell8 | ICELL8 (version 3) [ICELL8, Takara Bio] | 11 | 14 | [7b] |
| icell8-5-prime | ICELL8 (version 3) [ICELL8, Takara Bio] | 10 | N/A | [7b] |
| icell8-full-length | ICELL8 (version 3) [ICELL8, Takara Bio] | 16 | N/A | [7b] |
| indrops-v1[d,e] | inDrop (version 1) | 19 | 6 | [25,26] |
| indrops-v2[d,e] | inDrop (version 2) [Vendor: 1CellBio[f]] | 19 | 6 | [25,27] |
| indrops-v3[e,g] | inDrop (version 3) | 16 | 6 | [27] |
| nadia | Nadia [Nadia, Dolomite Bio] | 12 | 8 | |
| marsseq-v1 | MARS-Seq | 6 | 10 | [28] |
| marsseq-v2 | MARS-Seq 2.0 | 7 | 8 | [29] |
| microwell | Microwell-Seq | 18 | 6 | [30] |
| quartz-seq | Quartz-Seq | 6 | N/A | [31] |
| quartz-seq2-1536 | Quartz-Seq2 (1536 wells) | 15 | 8 | [32] |
| quartz-seq2-384 | Quartz-Seq2 (384 wells) | 14 | 8 | [32] |
| ramda-seq | RamDA-Seq | 6 | N/A | [33] |
| sciseq2[c,g] | SCI-seq (2-level indexing) | 30 | 8 | [1,34] |
| sciseq3[c,g] | SCI-seq (3-level indexing) | 40 | 8 | [1,34] |
| scifiseq | scifi-seq | 27 | 8 | [35] |
| scrbseq | SCRB-Seq, mcSCRB-Seq | 6 | 10 | [36,37] |
| seqwell | plexWell [Vendor: seqWell] | 12 | 8 | [38] |
| smartseq | SMART-Seq (version 1) | 16 | N/A | [39] |
| smartseq2 | SMART-Seq (version 2) [Vendor: Takara Bio] | 16 | N/A | [40] |
| smartseq2-UMI | SMART-Seq (version 2) | 16 | 8 | [8,10b] |
| smartseq3 | SMART-Seq (version 3) | 16 | 8 | [8,10b] |
| splitseq[c,d,h] | SPLiT-Seq [Vendor: Parse BioSciences] | 18 | 10 | [41] |
| strt-seq | STRT-Seq | 6 | N/A | [42] |
| strt-seq-c1 | STRT-Seq-C1 | 8 | 5 | [43] |
| strt-seq-2i | STRT-Seq-2i | 13 | 6 | [44] |
| surecell[h] | SureCell [ddSEQ, Bio-Rad] | 18 | 8 | [45,46] |

[a]Barcode length is max or total (linkers are removed automatically where needed) excluding barcodes in the index files which requires demultiplexing.
[b]Test data used in our study was generated from data this paper originally published.
[c]These technologies have their UMIs before their barcodes. The positions of UMIs and barcodes are automatically inverted when these technologies are selected as options.
[d]These technologies have their barcodes and UMIs in R2 rather than R1. The functional roles of R1 and R2 are automatically inverted when these technologies are selected as options.
[e]These technologies have their barcodes in two segments with Barcode-1 (8-11 bp) and Barcode-2 (8 bp). A barcode of 19 bp of the adjusted R1 file is used by filling in missing values with linker sequences.
[f]Vendor has since declared bankruptcy: http://1cellbio.com (Accessed April 1, 2020).
[g]These technologies have dual indexes (I1 and I2 from the i7 and i5 indexes from Illumina), which contain additional information on cell barcodes.
[h]These technologies have their barcodes in three segments with Barcode-1 (6 bp), Barcode-2 (6 bp), and Barcode-3 (6 bp). The first 18 bp of the adjusted R1 file is originally recognized as the cell barcode.

## Discussion

With the availability of a Docker image and GUI application for UniverSC, we envision UniverSC will facilitate robust and user-friendly single-cell analysis to democratise scRNA-seq technologies. As single-cell technologies become integral to a wide range of studies, mitigation of technical errors and integration of scRNA-seq data generated across different groups and platforms will be necessary. Processing data that contains various barcode and UMI configurations under a consistent framework will be convenient and essential. While there are pipelines that can be configured for a variety of technologies (dropSeqPipe[9]; zUMIs[10]; dropEst[14], Kallisto/BUStools[15]), Cell Ranger performs well in a server or cluster environment and generates a rich

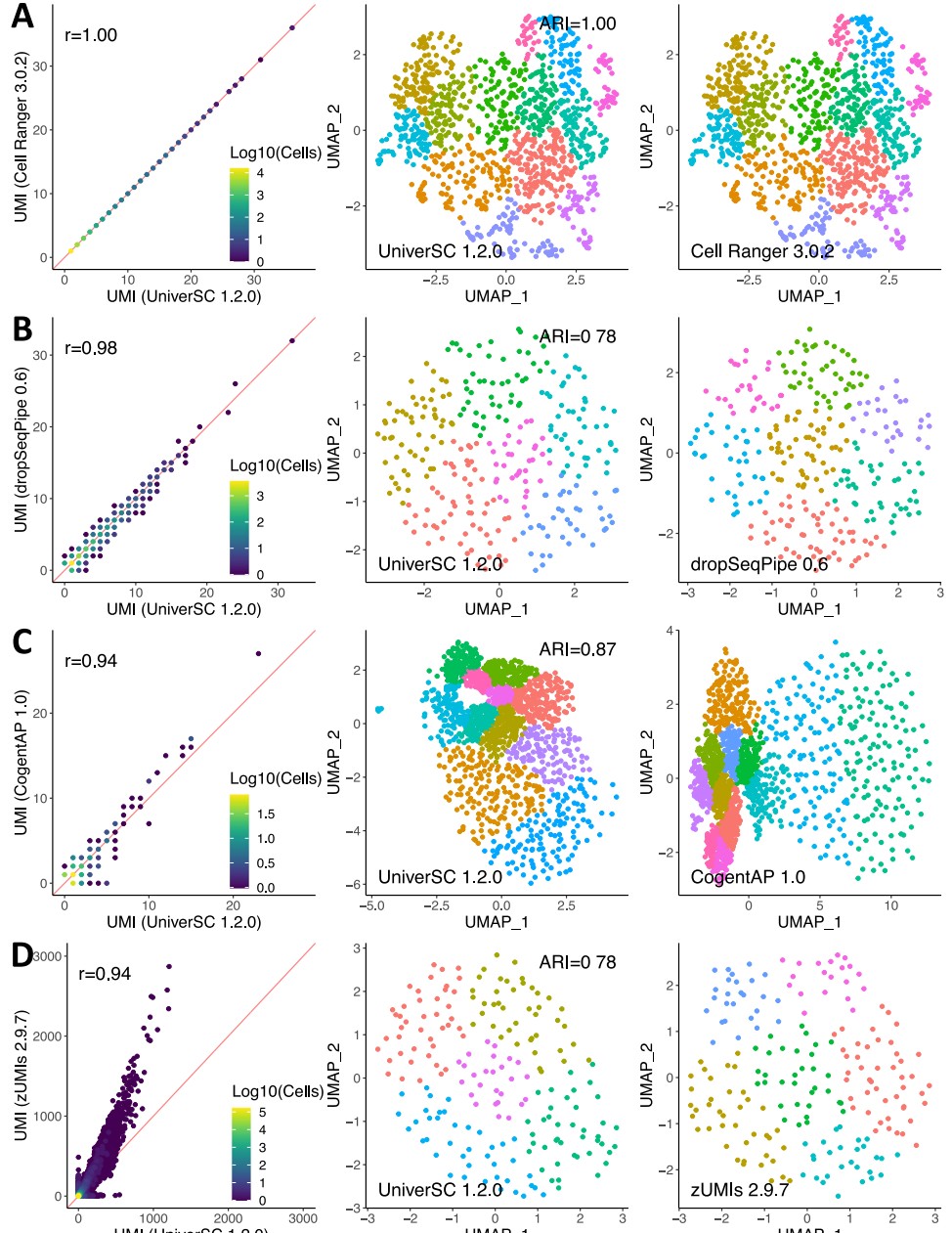

**Fig. 2 | Similarity assessment of UniverSC against other pipelines.** Comparisons between pairs of GBMs generated by UniverSC 1.2.0 against Cell Ranger 3.0.2 (**A**), dropSeqPipe 0.6 (**B**), CogentAP 1.0 (**C**), and zUMIs 2.9.7 (**D**). Direct comparison of GBMs, i.e., comparison of UMI counts for each gene for each cell, is on the left column. When data points (cells) overlap, the extent of data points overlapping is indicated by the colour of the dot from blue (no overlap) to yellow (most overlap). This is followed by the clustering results from UniverSC 1.2.0 (centre column) and its counterpart (right column), in which clusters are represented by colours. Source data is provided as a Source Data file.

and informative output summary. It is of note that UniverSC utilises Cell Ranger version 3.0.2 due to licensing. Although later versions of Cell Ranger are now available, since core changes enable analyses other than scRNA-seq, such as scATAC-seq, TCR, and BCR analyses, these updates do not majorly affect scRNA-seq data processing. As novel single-cell technologies are developed, the utility of UniverSC eliminates the need to develop a dedicated data processing pipeline for their own technology. Lastly, it will enable a fair comparison when evaluating the best platform for a specific sample type, which may be especially important with challenging samples, such as those containing large cells or digestive enzymes. We provide this tool for free and open-source to democratise single-cell analysis in a wide range of scientific applications.

## Methods

The set of input parameters for UniverSC is similar to that required by Cell Ranger, with a few additions. The UniverSC workflow requires paired-end FASTQ input files and reference data as prepared by Cell Ranger. By default, UniverSC assumes Read 1 of the FASTQ to contain the cell barcode and UMI and Read 2 to contain the transcript sequences which will be mapped to the reference, as is common in 3′ scRNA-seq protocols. Given a known barcode and UMI length, UniverSC will check the file name and barcodes, altering the configurations to match that of Chromium as needed. The chemistry appropriate for each single-cell technology for 3′ scRNA-seq is determined automatically (technologies for 5′ scRNA-seq other than that of Chromium are not supported at the time of writing). Data from

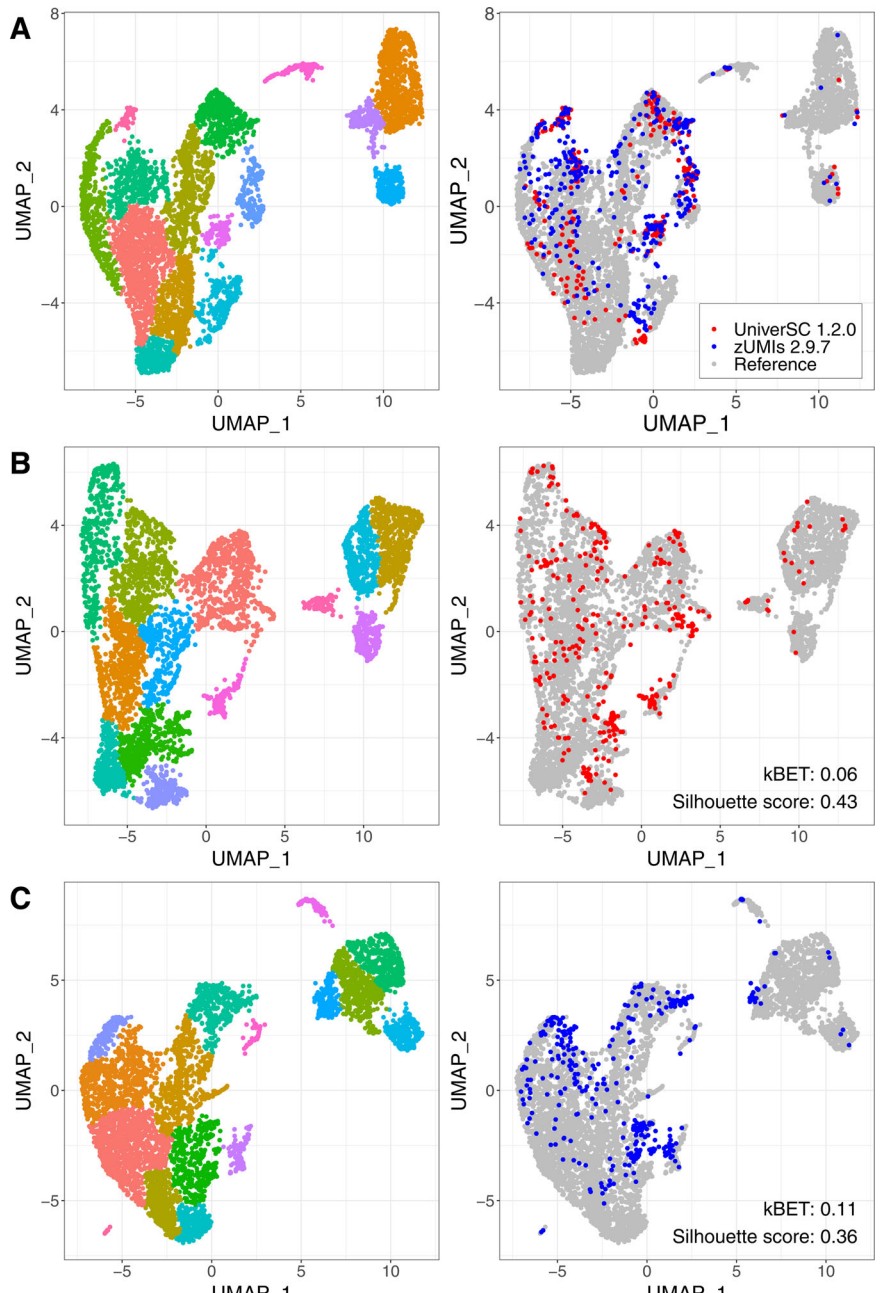

**Fig. 3 | Assessment of data integration by UniverSC versus multiple pipelines.** Integration output of Chromium data processed by UniverSC ("Reference" in grey), SmartSeq2 data processed via UniverSC (in red), and SmartSeq2 data processed via zUMIs (in blue). (**A**) A three-way integration. (**B**) Integration of reference and SmartSeq2 (UniverSC). (**C**) Integration of reference and SmartSeq2 (zUMIs). For (**B**) and (**C**), kBET and Silhouette scores are shown on the lower right corner. Different colours of dots represent different clusters in the left column. Source data is provided as a Source Data file.

multiple lanes is supported and so is using a custom set of barcodes specific to a given technology.

Published datasets of human cell lines were used to test for output similarity between UniverSC and other pipelines. Test datasets were prepared for Chromium[16], Drop-seq[6], ICELL8[7], and SmartSeq3[8,10] (see section Data availability for repositories and specific accession IDs for each dataset). The chromosome 21 (Chr21) of human genome GRCh38 (hg38) was used as the reference to process all datasets. For Chromium dataset, the 10x Genomics bamtofastq tool (https://github.com/10XGenomics/bamtofastq) was used to convert Cell Ranger 1.1.0 output from version 1 chemistry to be compatible with running newer versions. Only the reads that mapped to chromosome 21 were kept to reduce output data size. The ICELL8 dataset was further down sampled

to 250 K reads using seqtk[17] (sample with the same random seed for each read). Documentation and codes used to generate each filtered/downsized dataset are provided in the UniverSC GitHub repository (https://github.com/minoda-lab/universc). The output for UniverSC and the respective pipeline for each technology is provided as supplemental data (Supplementary Data 1–8). The full raw output is provided for Chromium, Drop-seq, and ICELL8 datasets. Only the processed GBM is provided for SmartSeq3 dataset due to the exceedingly large raw output size.

Each pair of raw GBMs, which is the critical portion of the pipeline output, were processed in parallel. The pair of GBMs was adjusted to have matching sets of barcodes and genes: only barcodes found in both GBMs were kept, and genes only found in one GBM were added to

the other with 0 UMIs assigned. The adjusted pair of GBMs was then used to carry out clustering analysis with an R package Seurat (version 4.1.1)[18] within R (version 4.1.2). Finally, the Pearson correlation between the GBMs and ARI between the two clustering outcomes were calculated using R packages stats (version 4.1.2) and clues (version 0.6.2.2) within R, respectively. For the scatterplots and computing Pearson correlation, a pair of UMI counts for each gene for each cell was considered a single data point unless they were both zero, e.g., up to 1000 data points would be compared for correlation for a pair of GBMs with 10 cells and 100 genes.

To demonstrate improvement on data integration, published datasets of mouse primary cells from a scRNA-seq benchmarking study were used[11]. From this study, we chose a dataset generated via Chromium as a reference and a dataset generated via SmartSeq2 as a comparison. The full mouse genome (GRCm39) was used as the reference genome and no downsampling was performed for these datasets. The reference Chromium dataset was processed once by UniverSC to generate one GBM, and the SmartSeq2 dataset was processed twice, once by UniverSC and once by zUMIs, to generate a pair of GBMs. The two SmartSeq2 GBMs were formatted as described above to have identical genes and barcodes. All three GBMs were formatted as described above to have identical genes (but not barcodes). Then each version of SmartSeq2 GBM was integrated with the reference GBMs independently using Seurat. To evaluate the quality of integration, kBET[12] and Silhouette score[13] were calculated for each case using R packages kBET (version 0.99.6) and cluster (version 2.1.3), respectively. The 3 output GBMs are provided as supplemental data (Supplementary Data 9–11).

We provide documentation for UniverSC accessible as a manual and help system in the terminal and a user-interface, which checks file inputs and gives error messages to identify potential problems. UniverSC can be run on any Unix-based system in the shell and both the source code and a docker image are publicly available (see Code availability). The user can also choose to install a GUI for UniverSC (see Code availability). We recommend installing UniverSC in a local directory (e.g., to a home directory) or somewhere appropriate with write access; it can be run on any system with Cell Ranger installed (i.e., added to the PATH environment variable). We also recommend running UniverSC on a server with sufficient memory to run the STAR alignment algorithm. Submission to a cluster in parallel with a job scheduler is supported but note that UniverSC can only run on one technology at a time due to the different barcode whitelist requirements. See the manual for further details. Note that UniverSC was developed by a third-party unrelated to 10x Genomics, and the most recent open-source version of Cell Ranger (version 3.0.2) is used with Cloupe (a portion of Cell Ranger) inactivated to comply with the 10x Genomics End User Software Licence Agreement.

## Data availability

The Chromium[16] (HEK293T human kidney cell-lines) dataset used in this study is available from the 10x Genomics website (10x Genomics: https://www.10xgenomics.com). The Drop-seq[6] dataset used in this study is available from GEO under accession code GSE63473. The ICELL8[7] dataset used in this study is available from EGA under accession code EGAD00001003443. SmartSeq3 dataset used in this study is available from EMBL ArrayExpress under accession code E-MTAB-8735. Chromium[16] and SmartSeq2[8,10] datasets used in this study for data integration test are both available from GEO under accession code GSE133549. All previously published datasets are under no restrictive access except for the ICELL8 dataset. To access the ICELL8 dataset, please consult the Genentech Data Access Committee as described in the aforementioned link. Source data are provided with this paper.

## Code availability

The most recent source code for UniverSC is publicly available along with installation instructions at GitHub (https://github.com/minoda-lab/universc) and the specific version of UniverSC used to generate data for this study is available at Zenodo[19]. The Docker image at DockerHub with all dependencies installed from source (https://hub.docker.com/repository/docker/tomkellygenetics/universc). We also provide a cross-platform application to run UniverSC via a GUI, available for macOS, Windows, and Linux Ubuntu (https://genomec.gsc.riken.jp/gerg/UniverSC). This comes along with a step-by-step installation and usage guide at (https://genomec.gsc.riken.jp/gerg/UniverSC/UniverSC_app_release/).

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

## Acknowledgements

This work was supported by a JSPS KAKENHI Grant-in-Aid for Scientific Research on Innovative Areas "Principles of pluripotent stem cells underlying plant vitality" (JP17H06470 to AM and 17H06472 to MH) and Center for IMS. We acknowledge contributions from Tommy Terooatea (RIKEN IMS) for testing UniverSC, Jonathon Moody and Chung-Chau Hon (RIKEN IMS) for their insightful discussions. We thank Musa Mhlanga (RIMLS) for encouraging this tool to be published. We also wish to acknowledge Shuwen Chen, Tsuyoshi Okumo, Max Sanchez, and Karthik Swaminathan (Takara Bio) for supporting data analysis from the ICELL8 platform with their CogentAP pipeline. We thank the developers at 10x Genomics of Cell Ranger and dependencies for making their code publicly available. We also thank Marcus Kinsella (CZI) for releasing a docker image of an open-source version of Cell Ranger 2.0.2.

## Author contributions

S.T.K. and K.B. conceptualised and wrote the UniverSC script, carried out the comparative analysis, and wrote the manuscript. S.T.K. documented the code and built the Docker image. R.A. developed the UniverSC GUI application and app documentation. N.A.H. generated datasets and tested the script. M.H. and A.M. supervised the project. A.M. edited the manuscript.

## Competing interests

The authors declare no competing interests.
