## [Transparent Peer Review File · Nature Communications]

A flexible cross-platform single-cell data processing pipelineREVIEWER COMMENTS

Reviewer #1 (Remarks to the Author: Overall significance):

The authors have developed UniverSC, a wrapper for the 10X Genomics CellRanger software that is compatible with 19 different single cell technologies. UniverSC modifies the cell barcode and Unique Molecular Index (UMI) to be compatible with CellRanger, thus enabling users to generate gene expression matrices from a variety of single cell technologies. The authors demonstrate UniverSC on datasets generated using 10X genomics, Drop-seq, and ICELL8 technologies and benchmark UniverSC results against existing pipelines for those technologies.

As single cell sequencing becomes cheaper and more popular, democratizing and simplifying the analysis is critical. I applaud the authors for attempting to generate a tool that can easily enable analysis across single cell technologies. However, while the concept of UniverSC certainly has broad potential, I'm not sure that the UniverSC tool as currently implemented makes single cell analysis significantly easier over existing tools such as DropEst or Kallisto/Bustools, which are other single cell analysis pipelines that can be configured to work with multiple single cell technologies.

Reviewer #1 (Remarks to the Author: Impact):

As it currently stands, I don't think UniverSC has a significant impact over existing tools for processing single cell RNA-seq data from multiple technologies such as Kallisto/Bustools and DropEst. While UniverSC has pre-configured settings for each technology, installing and running UniverSC still requires some amount of familiarity with Unix systems. I'm not sure it really is that much more effort to set the configurations for DropEst or Kallisto/Bustools vs using UniverSC. Additionally, while it was great to see a Docker container provided for users who want to run analyses on their laptops or home computers, the fact that UniverSC uses CellRanger means that most users can't realistically use UniverSC on their laptops or home machines since CellRanger has a pretty large memory requirement (32 gb RAM) and is quite slow.

Reviewer #1 (Remarks to the Author: Strength of the claims):

Critical improvements

1. I think a critical improvement to UniverSC could be adapting it to work with not only CellRanger, but also DropEst and Kallisto/Bustools, each of which improves on CellRanger in a number of ways. Giving users the option to use any of these three analysis pipelines across a variety of single cell technologies would enable UniverSC to be a powerful, universal platform for analysis. I think adapting UniverSC to work with Kallisto/Bustools, or another lightweight pipeline like Salmon-Alevin-fry, is especially critical as these methods have much smaller memory footprints and are much faster than CellRanger, enabling users to realistically process single cell data on their laptops or home machines.

2. Another critical improvement is making sure UniverSC is compatible with combinatorial indexing technologies such as Split-Seq and sci-RNA-seq. While the authors state that they are currently working on incorporating these technologies,

ensuring that UniverSC works with these technologies before publication is important. A commercially produced Split-Seq kit has recently been released and it seems likely that a large fraction of future single cell datasets will be generated with these combinatorial indexing technologies.

Optional suggestions

1. Making a graphical user interface for UniverSC would do quite a bit to enable scientists who are not familiar with Unix systems to process single cell datasets. However, I understand that making GUIs can be extremely time consuming.
2. Demonstrating the unique utility of UniverSC in a main figure would be really helpful. For example, if the authors could show that by re-processing all of these datasets with UniverSC, they were able to reduce the strength of technology specific batch effects, that would be a more convincing demonstration of UniverSC's impact.

Reviewer #1 (Remarks to the Author: Reproducibility):

The authors provide an easy to use shell script and Docker container for reproducibility, which is great to see.

Some minor comments on the figure layout and reporting summary:

1. The way Figure 2 is currently laid out is a bit confusing. I'm currently not sure how to interpret the plots in each column as the descriptions in the figure legends are a bit sparse. I think it would be helpful to arrange the panels so that each column is a panel, enabling a more thorough description of each type of plot in the figure legends.
2. I'm also not sure I understand the correlation plot in the first column of Figure 2. If this is a direct comparison of gene expression matrices, why are there so few data points?
3. The github link in the reporting summary is broken

Reviewer #2 (Remarks to the Author: Overall significance):

Kelly et al., present a pipeline called UniverSC that aims to wrap preprocessing pipelines from several single-cell protocols in a single scripted tool. UniverSC is mainly wrapping configurations files around the CellRanger pipeline, so that multiple protocols are handled.

Reviewer #2 (Remarks to the Author: Impact):

Despite its potential utility for sequencing and bioinformatics service facilities,

UniverSC does not show any novelty, and is a simple wrapper around existing tools. Moreover, the authors don't mention that STARsolo can also perform these tasks, without the need for the heavy CellRanger pipeline. The manuscript seems therefore of very limited interest.

Reviewer #2 (Remarks to the Author: Strength of the claims):

N/A

Reviewer #2 (Remarks to the Author: Reproducibility):

N/A

Reviewer #3 (Remarks to the Author: Overall significance):

Kelly et al. developed a universal single-cell data processing tool named UniverSC, which could support any UMI-based platform. UniverSC may be useful for the integration, comparison, and evaluation across data generated from different platforms.

Reviewer #3 (Remarks to the Author: Impact):

UniverSC could benefit the single-cell sequencing data analysis.

Reviewer #3 (Remarks to the Author: Strength of the claims):

UniverSC could be a useful tool for single-cell sequencing data processing. I have the following concerns:

1. The authors only tested the performance of UniverSC on three datasets. If more datasets could be tested, their results/conclusions will be more solid.
2. Does UniverSC have limit in the number of cells for processing? If yes, what is the largest number of cells can be analyzed?
3. What is the speed for UniverSC handling the single-cell sequencing data?
4. How much memory does UniverSC need to process the single-cell sequencing data with 10,000 cells?

Reviewer #3 (Remarks to the Author: Reproducibility):

Could the authors provide more details about how they process the test datasets in the Methods section? For example, how many cells does each dataset have? Did they filter any cells or use all the cells of each dataset in the analysis?

REVIEWER COMMENTS

Reviewer #1 (Remarks to the Author: Overall significance):

The authors have developed UniverSC, a wrapper for the 10X Genomics CellRanger software that is compatible with 19 different single cell technologies. UniverSC modifies the cell barcode and Unique Molecular Index (UMI) to be compatible with CellRanger, thus enabling users to generate gene expression matrices from a variety of single cell technologies. The authors demonstrate UniverSC on datasets generated using 10X genomics, Drop-seq, and ICELL8 technologies and benchmark UniverSC results against existing pipelines for those technologies.

As single cell sequencing becomes cheaper and more popular, democratizing and simplifying the analysis is critical. I applaud the authors for attempting to generate a tool that can easily enable analysis across single cell technologies. However, while the concept of UniverSC certainly has broad potential, I'm not sure that the UniverSC tool as currently implemented makes single cell analysis significantly easier over existing tools such as DropEst or Kallisto/Bustools, which are other single cell analysis pipelines that can be configured to work with multiple single cell technologies.

We thank the reviewer for their kind comments and thorough review. We are glad the reviewer recognises the significance of our work for the wider research community. As described below in more detail, we have taken Reviewer 1's suggestion and developed a GUI that runs UniverSC for Mac, Linux, and Windows. We trust that this will make conducting single cell analysis significantly easier for a wider community, especially for non-bioinformaticians, democratising single cell RNA-seq technology.

Reviewer #1 (Remarks to the Author: Impact):

As it currently stands, I don't think UniverSC has a significant impact over existing tools for processing single cell RNA-seq data from multiple technologies such as Kallisto/Bustools and DropEst. While UniverSC has pre-configured settings for each technology, installing and running UniverSC still requires some amount of familiarity with Unix systems. I'm not sure it really is that much more effort to set the configurations for DropEst or Kallisto/Bustools vs using UniverSC. Additionally, while it was great to see a Docker container provided for users who want to run analyses on their laptops or home computers, the fact that UniverSC uses CellRanger means that most users can't realistically use UniverSC on their laptops or home machines since CellRanger has a pretty large memory requirement (32 gb RAM) and is quite slow.

Reviewer #1 (Remarks to the Author: Strength of the claims):

Critical improvements

1. I think a critical improvement to UniverSC could be adapting it to work with not only CellRanger, but also DropEst and Kallisto/Bustools, each of which

improves on CellRanger in a number of ways. Giving users the option to use any of these three analysis pipelines across a variety of single cell technologies would enable UniverSC to be a powerful, universal platform for analysis. I think adapting UniverSC to work with Kallisto/Bustools, or another lightweight pipeline like Salmon-Alevin-fry, is especially critical as these methods have much smaller memory footprints and are much faster than CellRanger, enabling users to realistically process single cell data on their laptops or home machines.

We agree that giving users more options is beneficial to the wider research community. However, we are not sure if there would be significant added benefits in adapting UniverSC to support other pipelines such as DropEst or Kallisto/Bustools for two reasons. 1) We believe that having fewer dependencies and fewer choices for a novice user can avoid confusion. 2) dropEst, dropSeqPipe, and Kallisto/Bustools already support multiple technologies, but Cell Ranger does not, and UniverSC opens up Cell Ranger for other technologies.

The computational requirements for Cell Ranger may be higher than other pipelines, thus for large computations we recommend running UniverSC on a cluster environment in parallel. That being said, the requirements for Cell Ranger given by 10x Genomics are specific to their whitelist of over 3 million barcodes (for 3' scRNA v3) on a 3.2 Gbp human genome. UniverSC can still be executed on a laptop for small genomes, barcode whitelists, or MiSeq datasets. In our experience, ICELL8 samples can be run easily on a laptop. Additionally, many laboratories attempting to conduct scRNAseq have a dedicated desktop computer for bioinformatics powerful enough to handle such tasks.

2. Another critical improvement is making sure UniverSC is compatible with combinatorial indexing technologies such as Split-Seq and sci-RNA-seq. While the authors state that they are currently working on incorporating these technologies, ensuring that UniverSC works with these technologies before publication is important. A commercially produced Split-Seq kit has recently been released and it seems likely that a large fraction of future single cell datasets will be generated with these combinatorial indexing technologies.

We agree that combinatorial indexing has great potential and that supporting commercially available technologies is very important as these will be widely available to labs. UniverSC has been updated since the initial submission and now fully supports combinatorial indexing technologies including sci-RNA-seq and SPLiT-Seq (which has recently been commercialized). With regards to commercially available technologies, UniverSC now supports BD Rhapsody (Becton, Dickinson and Company), C1 (Fluidigm), Chromium (10x Genomics), ICELL8 (Takara Bio), inDrops (1CellBio), Nadia (Dolomite Bio), PlexWell (SeqWell), SPLiT-Seq (Parse Biosciences), SmartSeq (Takara Bio), and SureCell/ddSEQ (BioRad). In addition, the new version of UniverSC also supports 5' scRNA with SmartSeq3 and dual indexes for inDrops v3.

Test data for SmartSeq3 is now included with UniverSC, and we have added it in our manuscript, comparing SmartSeq3 data processing via UniverSC and zUMI (pipeline originally used to analyze SmartSeq3 data)

Optional suggestions

1. Making a graphical user interface for UniverSC would do quite a bit to enable scientists who are not familiar with Unix systems to process single cell datasets. However, I understand that making GUIs can be extremely time consuming.

Thank you for this suggestion. We completely agree with this thought and decided to take up the challenge of building a GUI for UniverSC as a major revision of this manuscript. To this end, we have added a new author, Radu Abu Ras, who developed the GUI. This application runs UniverSC on Linux, Mac OS, and Windows (a Docker image is needed for Mac OS and Windows). Running UniverSC on a GUI app through Docker minimizes the use of command-line, which is an obvious benefit for those who are not familiar with command lines. We have tested and ensured that it is functional on all 3 operating systems. https://genomec.gsc.riken.jp/gerg/UniverSC/UniverSC_Release/ This revision is reflected in line XXX and XXX in the manuscript.

2. Demonstrating the unique utility of UniverSC in a main figure would be really helpful. For example, if the authors could show that by re-processing all of these datasets with UniverSC, they were able to reduce the strength of technology specific batch effects, that would be a more convincing demonstration of UniverSC's impact.

We thank the reviewer for raising this thoughtful point but demonstrating this is out of scope of the current work. We provide small test datasets derived from published data to demonstrate (1) that the software functions as documented and (2) that the outcome is similar but not identical. We believe we have sufficiently shown that different pipelines can generate different outcomes from identical inputs and simply eliminating this would result in less bias upon the analyses further downstream.

Reviewer #1 (Remarks to the Author: Reproducibility):

The authors provide an easy-to-use shell script and Docker container for reproducibility, which is great to see.

Some minor comments on the figure layout and reporting summary:

1. The way Figure 2 is currently laid out is a bit confusing. I'm currently not sure how to interpret the plots in each column as the descriptions in the figure legends are a bit sparse. I think it would be helpful to arrange the panels so that each column is a panel, enabling a more thorough description of each type of plot in the figure legends.

Please see the response to the next comment.

2. I'm also not sure I understand the correlation plot in the first column of Figure 2. If this is a direct comparison of gene expression matrices, why are there so few data points?

We clarified the description of the figure. We assume the confusion by the reviewer is coming from the left column in Figure 2. In short, the UMI counts for each gene in each cell is compared between the GBMs generated by the two data processing pipelines being compared. So, if the GBM had 10 cells (10 columns) and 100 genes (100 rows) there would be 1000 data points on the plot. Overlaid points are coloured as shown in the key to avoid obstruction.

3. The github link in the reporting summary is broken

The manuscript has been updated to correct the GitHub URL.

From: <https://github.com/minodalab/universc>

To: <https://github.com/minoda-lab/universc>

Reviewer #2 (Remarks to the Author: Overall significance):

Kelly et al., present a pipeline called UniverSC that aims to wrap preprocessing pipelines from several single-cell protocols in a single scripted tool. UniverSC is mainly wrapping configuration files around the CellRanger pipeline, so that multiple protocols are handled.

Reviewer #2 (Remarks to the Author: Impact):

Despite its potential utility for sequencing and bioinformatics service facilities, UniverSC does not show any novelty, and is a simple wrapper around existing tools. Moreover, the authors don't mention that STARsolo can also perform these tasks, without the need for the heavy CellRanger pipeline. The manuscript seems therefore of very limited interest.

We respectfully disagree with the reviewer regarding the merits that UniverSC can offer. Despite it being specifically designed for datasets generated by 10x platform, Cell Ranger has gained its popularity at least partly due to its rich summary data. We believe that allowing the users to access the same summary data for datasets generated by other platforms would be beneficial for the wider research community and facilitate integration of multi-platform studies. The authors are aware of the advantages of STARsolo but did not cite this technique in the manuscript as it had not been published at the time of submission.

Reviewer #2 (Remarks to the Author: Strength of the claims):

N/A

Reviewer #2 (Remarks to the Author: Reproducibility):

N/A

Reviewer #3 (Remarks to the Author: Overall significance):

Kelly et al. developed a universal single-cell data processing tool named UniverSC, which could support any UMI-based platform. UniverSC may be

useful for the integration, comparison, and evaluation across data generated from different platforms.

Reviewer #3 (Remarks to the Author: Impact):

UniverSC could benefit the single-cell sequencing data analysis.

The authors agree wholeheartedly.

Reviewer #3 (Remarks to the Author: Strength of the claims):

UniverSC could be a useful tool for single-cell sequencing data processing. I have the following concerns:

1. The authors only tested the performance of UniverSC on three datasets. If more datasets could be tested, their results/conclusions will be more solid.

Per reviewer's suggestion, we have also added and tested SmartSeq3 (full-length) datasets. Comparison of outputs from UniverSC and zUMI (the data processing tool developed for Smart-Seq3 is reflected in Figure 3, line 104-118 in the revised manuscript.

2. Does UniverSC have limit in the number of cells for processing? If yes, what is the largest number of cells can be analyzed?

These limitations will depend on the computational resources available to the researcher. In principle it can be run on very large datasets and the memory requirements depend on Cell Ranger as described in the documentation for this dependency.

3. What is the speed for UniverSC handling the single-cell sequencing data?

As UniverSC is a wrapper for Cell ranger Version 3.0.2, the speed of UniverSC is nearly equal to the speed of Cell Ranger.

4. How much memory does UniverSC need to process the single-cell sequencing data with 10,000 cells?

Again, the minimum requirement to run Cell ranger Version 3.0.2 is 16 GB as this is the minimum requirement for STAR aligner, which UniverSC (Cell Ranger) depends on.

Reviewer #3 (Remarks to the Author: Reproducibility):

Could the authors provide more details about how they process the test datasets in the Methods section? For example, how many cells does each dataset have? Did they filter any cells or use all the cells of each dataset in the analysis?

The raw published data was downloaded from the respective repository for each technique and subsampled to create a small test dataset. To ensure reproducibility

the steps necessary to do this are documented with the files and all scripts used to do it are included with UniverSC. It should not be necessary to filter the data to reproduce our results as we provide the filtered test datasets in the `universc/test/` directory and the outputs for UniverSC, Cell Ranger, dropSeqPipe, Cogent AP, and zUMI are provided as supplementary data. As described in the documentation and manuscript, these datasets are provided for technical testing purposes only and are not indicative of biological results. Therefore, no changes are needed to the manuscript to ensure reproducible results.

Reviewer comments:

Reviewer #1 (Remarks to the Author: Overall significance):

The authors have updated UniverSC, a wrapper for the 10X Genomics CellRanger software that is compatible with 19 different single cell technologies. UniverSC modifies the cell barcode and Unique Molecular Index (UMI) to be compatible with CellRanger, thus enabling users to generate gene expression matrices from a variety of single cell technologies. The authors have added a GUI that enables Windows/Mac users to run UniverSC with minimal command line expertise necessary. The authors have also added support for combinatorial indexing technologies such as Split-Seq.

Reviewer #1 (Remarks to the Author: Impact):

A simple command line tool that can process datasets from multiple single cell RNA-seq technologies is very convenient, although not necessarily groundbreaking and I can see a fairly substantial user base for this tool in both academia and industry. The ability to process single-cell RNA-seq data via a GUI (graphical user interface) on local Windows/Mac is especially interesting since it democratizes single-cell analysis. However, there are still some issues with both the demonstrated examples and the GUI interface that I'll go into in more detail in the next 2 sections.

Reviewer #1 (Remarks to the Author: Strength of the claims):

The addition of the GUI is great, I have some comments about the documentation and the overall protocol in the next section.

Critical improvements:

While I understand this is primarily a methods paper, the benchmarking analysis of UniverSC seems a bit shallow to me. The authors compare UniverSC with CellRanger, zUMI, and other processing tools (and show that UniverSC exactly corresponds with CellRanger as it should). However, I'm not sure how much these comparisons really demonstrate on their own. One thing I'd like to see is an example of how processing datasets from different technologies using UniverSC can help with dataset integration. For example, processing a 10X dataset and a SmartSeq3 dataset of the same cell types with UniverSC should result in better dataset integration than if the 10X dataset was processed with CellRanger and the SmartSeq3 dataset with zUMI. This should be fairly straightforward given that the authors already showed that the correlation between UniverSC/CellRanger and another technique, zUMI is fairly low to due differences in counting multimapping reads.

Minor comments:

While the GUI is great, UniverSC/CellRanger is incredibly memory intensive and slow which strains laptop processing capabilities. I understand the complexities involved in adding additional tools to UniverSC but I do think making UniverSC less computationally intensive is critical for widespread adoption.

The authors clarified that the correlation plots in the main figures are actually density hex plots in the captions which is great. However, this seems somewhat unnecessarily complex and a standard scatterplot might be a bit less confusing.

Reviewer #1 (Remarks to the Author: Reproducibility):

I tried out the GUI on my Windows laptop and while the design was impressive, the documentation needs quite a bit of work. Given that most users who would try a GUI have very little command line expertise or experience with tools like docker, there needs to be a step by step guide on how to install docker and pull the UniverSC image. It looks like the only way to pull a docker image on Windows is through the Windows Terminal so that part definitely needs to be explicit in the tutorial.

It would also be great if in addition to the standalone GUI binary, I could download a single folder with the GUI, a pre-built human or mouse reference (since these will most likely be the most common ones), and a small test dataset. While these files are all available separately, it would greatly streamline UniverSC if they were available together for the most common use cases.

Reviewer #3 (Remarks to the Author: Overall significance):

The authors have addressed my concerns.

REVIEWER COMMENTS

Reviewer #1 (Remarks to the Author: Overall significance):

The authors have updated UniverSC, a wrapper for the 10X Genomics, CellRanger software that is compatible with 19 different single cell technologies. UniverSC modifies the cell barcode and Unique Molecular Index (UMI) to be compatible with CellRanger, thus enabling users to generate gene expression matrices from a variety of single cell technologies. The authors have added a GUI that enables Windows/Mac users to run UniverSC with minimal command line expertise necessary. The authors have also added support for combinatorial indexing technologies such as Split-Seq.

We again thank the reviewer for the thorough review. We are glad the reviewer recognizes the improvements we have made from our previous submission. We have taken Reviewer 1's suggestions and made improvements accordingly. As an additional note, UniverSC is now compatible with 40 different technologies.

Reviewer #1 (Remarks to the Author: Impact):

A simple command line tool that can process datasets from multiple single cell RNA-seq technologies is very convenient, although not necessarily groundbreaking and I can see a fairly substantial user base for this tool in both academia and industry. The ability to process single-cell RNA-seq data via a GUI (graphical user interface) on local Windows/Mac is especially interesting since it democratizes single-cell analysis. However, there are still some issues with both the demonstrated examples and the GUI interface that I'll go into in more detail in the next 2 sections.

We are pleased that the reviewer recognizes the importance of this work and especially the cross-platform graphical user interface, which took considerable effort to develop. Please also note that the command-line interface has greater flexibility and allows for automation or use on a remote server environment.

Reviewer #1 (Remarks to the Author: Strength of the claims):

Critical improvements:

1. While I understand this is primarily a methods paper, the benchmarking analysis of UniverSC seems a bit shallow to me. The authors compare UniverSC with CellRanger, zUMI, and other processing tools (and show that UniverSC exactly corresponds with CellRanger as it should). However, I'm not sure how much these comparisons really demonstrate on their own. One thing I'd like to see is an example of how processing datasets from different technologies using UniverSC can help with dataset integration. For example, processing a 10X dataset and a SmartSeq3 dataset of the same cell types with UniverSC should result in better dataset integration than if the 10X dataset was processed with CellRanger and the SmartSeq3 dataset with zUMI. This should be fairly straightforward given that the authors already showed that the correlation between UniverSC/CellRanger and another technique, zUMI is fairly low to due differences in counting multimapping reads.

We thank the reviewer for the critical comment. We have made two major changes to the manuscript in light of this comment. One is that we now use a much larger dataset for SmartSeq3 than we did previously. The pattern that we saw previously with low correlation (-0.39) was likely an artifact of using a dataset of insufficient size. We now see that GBMs generated by UniverSC and zUMIs show a high correlation (0.94) but zUMIs generally had higher UMI counts, which is consistent with the fact that zUMIs counts multi-mapped reads whereas CellRanger does not.

We also agree with the reviewer's comment on demonstrating how integration of datasets processed by a single pipeline (i.e., UniverSC) would compare to when datasets are processed by different pipelines. We have carried this out and have added a new section to discuss this very point [lines 171-195 and lines 305-317].

Minor comments:

2. While the GUI is great, UniverSC/CellRanger is incredibly memory intensive and slow which strains laptop processing capabilities. I understand the complexities involved in adding additional tools to UniverSC but I do think making UniverSC less computationally intensive is critical for widespread adoption.

While we agree with the reviewer that such improvement would benefit many, since UniverSC is a simple wrapper for CellRanger, this is not possible unless we tamper with the CellRanger itself, which is beyond the scope of this manuscript. While we will always keep the door open for our future versions to incorporate such updates, we think that the current version of UniverSC is ready to be recognized as a publication. We have made the source code for UniverSC open-source and publicly available on a license that permits reuse. So, contributions from the research community to do this are welcome.

3. The authors clarified that the correlation plots in the main figures are actually density hex plots in the captions which is great. However, this seems somewhat unnecessarily complex and a standard scatter plot might be a bit less confusing.

In the case of Figures 2, we chose to use density hex instead of a simpler standard scatter plot because it can better reflect the major difference in the number of occurrences between different data points. All non-zero UMI pairs were considered upon calculating the correlation between two gene-barcode matrices. Since most UMI counts in these matrices are "0" and the occurrences generally decrease as the UMI count increases, the data points in the figure near the point of origin are weighted more heavily compared to those further away. Due to the large number of similar data values, it is not possible to use a scatter plot without losing this information by only using a single color for all the data points as the points would be overlaid.

Reviewer #1 (Remarks to the Author: Reproducibility):

1. I tried out the GUI on my Windows laptop and while the design was impressive, the documentation needs quite a bit of work. Given that most users who would try a GUI have very little command line expertise or experience with tools like docker, there needs to be a step-by-step guide on how to install docker and pull the

UniverSC image. It looks like the only way to pull a docker image on Windows is through the Windows Terminal so that part definitely needs to be explicit in the tutorial.

Thank you for this suggestion. We agree with the reviewer's comment and we now include a step-by-step guide on installing UniverSC GUI for Mac, Windows, and Linux.

2. It would also be great if in addition to the standalone GUI binary, I could download a single folder with the GUI, a pre-built human or mouse reference (since these will most likely be the most common ones), and a small test dataset. While these files are all available separately, it would greatly streamline UniverSC if they were available together for the most common use cases.

With regards to a small test dataset, we agree with the reviewer's comment wholeheartedly, and we now added 3 small datasets (one for 10x, Nadia, and ICELL8) together with the GUI release at (<https://genomec.gsc.riken.jp/gerg/UniverSC>). For those that are using UniverSC through the command-line, test datasets are already available within the package under "universc/test/shared". With regards to a pre-made human/mouse genome reference, we did not include them due to its large file sizes. While we understand it would be convenient in some cases. This would also make downloading the software and updating it time-consuming and difficult for users. However, we do provide these references together with the GUI release. We note that other software releases in the field, such as Cell Ranger, provide reference data separately as well and users in the field are accustomed to this.

REVIEWERS' COMMENTS:

Reviewer #4 - Expert in clinical bioinformatics and computational single-cell genomics

Remarks to the Author: Overall significance:

The authors describe UniverSC, a wrapper to reformat a wide range of single cell sequencing protocols to work with cellranger. The package calls cellranger and outputs gene-barcode matrices which can be further processed by downstream analysis workflows. Moreover, UniverSC offers a GUI for simplified parameter setting and to ease the use for non-bioinformaticians.

Reviewer #4 (Remarks to the Author: Impact):

I appreciate the usefulness of a tool to unify different kinds of single cell data, and see the main application for improved integrative analysis, as batch effects resulting from differences in the initial raw processing are removed. The general idea is very good and could actually have an impact in the field, where integrative analysis across experiments is still often challenging.

However, I see some major drawbacks that will potentially limit the actual interest of the community, refer to the comments below.

Reviewer #4 (Remarks to the Author: Strength of the claims):

1. While I agree with the authors that Cellranger is widely used in the field and thus presents itself as a good choice, still it comes with the downside of only offering a very outdated open-source release. UniverSC used Cellranger 3.0.2, while the latest version is already at 7.0.1. Thus, major improvements in the software are not accessible to users if they apply UniverSC.

A good open-source alternative, e.g. as an additional mapper to select, would be STARsolo. This way users would have access to an up-to-date software that also generates gene-barcode matrices like cellranger (and can thus be similarly plugged into downstream analysis workflows).

2. I fear that the GUI will not be usable in a real-life analysis, due to the resource requirements of a typical cellranger run. Unless the GUI can be run on a compute cluster, I do not see how users will be able to run their analysis on without anyhow reverting back to the command-line, as a typical laptop does not have the capacity to analyse a typical e.g. 10x experiment in a reasonable time (if at all). Is it possible to output set all parameters in the GUI and to output the command for command line application?

Reviewer #4 (Remarks to the Author: Reproducibility):

In line with Reviewer #1 I feel that the quality of the GUI documentation can still be improved. While the installation instructions are illustrative including screenshots, the actual manual of the GUI is only a plain text file that is somewhat difficult to follow as it

needs to describe different options (e.g. "you will see 2 tabs "Build Script" and "Generate Reference File"). It would be much better if also this manual would contain screenshots of the GUI.

Comments from Reviewer #4

(Remarks to the Author: Overall significance)

The authors describe UniverSC, a wrapper to reformat a wide range of single cell sequencing protocols to work with cellranger. The package calls cellranger and outputs gene-barcode matrices which can be further processed by downstream analysis workflows. Moreover, UniverSC offers a GUI for simplified parameter setting and to ease the use for non-bioinformaticians.

(Remarks to the Author: Impact)

I appreciate the usefulness of a tool to unify different kinds of single cell data, and see the main application for improved integrative analysis, as batch effects resulting from differences in the initial raw processing are removed. The general idea is very good and could actually have an impact in the field, where integrative analysis across experiments is still often challenging.

However, I see some major drawbacks that will potentially limit the actual interest of the community, refer to the comments below.

(Remarks to the Author: Strength of the claims)

1. While I agree with the authors that Cellranger is widely used in the field and thus presents itself as a good choice, still it comes with the downside of only offering a very outdated open-source release. UniverSC used Cellranger 3.0.2, while the latest version is already at 7.0.1. Thus, major improvements in the software are not accessible to users if they apply UniverSC.

A good open-source alternative, e.g. as an additional mapper to select, would be STARsolo. This way users would have access to an up-to-date software that also generates gene-barcode matrices like cellranger (and can thus be similarly plugged into downstream analysis workflows).

We thank the reviewer for their overall positive evaluation of the manuscript and identifying its shortcomings and strengths. We largely agree with the evaluation though we do not see “major drawbacks” in the same light as the reviewer. We address this in more detail below.

As the reviewer points out, multiple major updates have been made to Cell Ranger as a whole since July, 2020 (release of version 4.0). However, core changes introduced in these updates are related to analyses other than scRNAseq, such as scATACseq, TCR, and BCR analyses. UniverSC essentially runs “cellranger count” command of Cell Ranger for scRNAseq datasets, and the updates since version 3.0.2 do not majorly affect this command. According to release notes from 10x Genomics, main changes that potentially affect the outcome of “cellranger count” include:

- *UMAP dimension reduction included in summary in addition to t-SNE projections. (v3.1)*
- *--target-panel option for targeted gene expression analysis along with other options coupled with it. (v4.0)*

- *automatic trimming of the template switch oligo (TSO) sequence and the poly-A sequence from the 3' and 5' end of Read-2 respectively prior to sequence alignment when analyzing 3' data. (v4.0)*
- *--include-introns option for counting intronic reads using 3' and 5' gene expression products. (v5.0)*

Of these changes, the first two do not affect the crucial gene-barcode matrix generation, and third we already address in the manuscript by highly recommending to trim the reads prior to running UniverSC (separate script to do this is already included in the UniverSC package). The handling of intronic reads is the only update with the potential to impact the outcome made in the last two years. Thus, Cell Ranger 3.0.2 is a stable version for scRNAseq with no major changes in the pipeline since then. scRNAseq remains to be one of the most sought-after applications of single cell technologies and we believe the community still benefits a tremendous amount from UniverSC. We have added this point in Discussion.

Suggestion to incorporate STARsolo was also made earlier by Reviewer #2. Then, we respectfully disagreed, rebutted and convinced the reviewer not to do so (at least for now). We believe that the popularity of Cell Ranger is at least partly due to its rich summary output data. We successfully argued then and argue now that allowing the users to access the same summary data for datasets generated by other platforms would be beneficial for the wider research community and facilitate integration of multi-platform studies.

2. I fear that the GUI will not be usable in a real-life analysis, due to the resource requirements of a typical cellranger run. Unless the GUI can be run on a compute cluster, I do not see how users will be able to run their analysis on without anyhow reverting back to the command-line, as a typical laptop does not have the capacity to analyse a typical e.g. 10x experiment in a reasonable time (if at all). Is it possible to output set all parameters in the GUI and to output the command for command line application?

We agree with the reviewer's suggestion that it is best for the users to be able to view the command about to be run for them to copy and paste elsewhere. We already have this function implemented as the "View Script" icon directly right of the "Generate Reference" or "Run Script" icon. We have revised our step-by-step user guide to clarify this (please see below).

We do think that UniverSC GUI will have a substantial range of usership. The most used resource for UniverSC is RAM and the largest memory sink for UniverSC is STAR aligner. STAR aligner takes 27GB of RAM for aligning a sample to the human genome (<https://doi.org/10.1093/bioinformatics/bts635>). So, a computer with 32GB of RAM can comfortably handle most samples through UniverSC. While we would agree that a laptop with 32GB of RAM is not the cheapest, a quick search on Amazon gives us options for approximately \$1,500 and some as low as \$1,000. This price would be even lower for desktops, and it is also not unusual for a lab to already have at least one computer that meets this

requirement. Of course, it would be cheaper to rent computer space from services such as AWS, but those who can comfortably maneuver through such services would not have considered using the GUI in the first place. All things considered; we believe UniverSC to be a simpler option for many notwithstanding the hardware requirements that the reviewer highlights.

(Remarks to the Author: Reproducibility):

In line with Reviewer #1 I feel that the quality of the GUI documentation can still be improved. While the installation instructions are illustrative including screenshots, the actual manual of the GUI is only a plain text file that is somewhat difficult to follow as it needs to describe different options (e.g. "you will see 2 tabs "Build Script" and "Generate Reference File"). It would be much better if also this manual would contain screenshots of the GUI.

We agree with the reviewer. We have revised the user guide to not only include how to install UniverSC GUI App, but to also include how to make a new reference and process samples. In which, we clearly mention the computational requirements for RAM and how to view the command that would be passed on.